# Introducing gold-standard essential gene datasets for *Pseudomonas aeruginosa* to enhance Tn-Seq analyses

Cléophée Van Maele[1☉], Ségolène Caboche[2☉*], Nathan Nicolau-Guillaumet[3], Anaëlle Muggeo[3], Thomas Guillard[iD][3*]

1 Université de Reims Champagne-Ardenne, INSERM, P3Cell, U 1250, Reims, France, 2 Université de Lille, CNRS, Inserm, CHU Lille, Institut Pasteur de Lille, US41-UAR 2014-PLBS, Lille, France, 3 Université de Reims Champagne-Ardenne, INSERM, CHU de Reims, Laboratoire de bactériologie-Virologie-Hygiène hospitalière, P3Cell, U 1250, Reims, France

☉ These authors contributed equally to this work.
* tguillard@chu-reims.fr (TG); segolene.caboche@univ-lille.fr (SC)

## Abstract

Transposon Sequencing (Tn-Seq) is a high-throughput technique that utilizes transposon mutant libraries to assess gene fitness or essentiality under specific conditions potentially identifying novel therapeutic targets. However, the diversity of statistical methods, bioinformatics tools, and parameters complicates the selection of the most appropriate and reliable analysis pipeline for a given dataset. A significant limitation of existing studies is the absence of a gold-standard set of essential genes (EGs) for evaluating the analysis process. Relying on the original study as a gold-standard is suboptimal, as these results may have been obtained using non-optimal tools. Here, we introduce reliable EG datasets for *Pseudomonas aeruginosa* to enhance Tn-Seq analyses. By utilizing literature data and sequencing of six samples from PA14 Wild-Type (WT) and PA14 OprD-deficient (Δ*oprD*), grown in LB medium, we compared EG lists generated by several statistical methods of TRANSIT2 and by the FiTnEss tools. We established a reference dataset of 84 genes found in *P. aeruginosa* and another gold-standard set composed of 115 genes specific to PA14 grown in LB. Our findings revealed that depending on the analysis method used, retrieval rates of gold-standard genes ranged from 0% to 100%. The Hidden-Markov Model (HMM) method available in TRANSIT2 identified approximately 90% of gold-standard EGs, while FiTnEss identified up to 100%. This study addressed a critical gap in the field by providing gold-standard sets of EGs, enabling comparative evaluation of Tn-Seq analysis methods to help researcher select the most suitable bioinformatics pipeline for a given Tn-Seq dataset. We anticipate that our results will facilitate Tn-Seq analysis comparisons, harmonize *P. aeruginosa*-related studies, promote standardization and enhance reproducibility. Ultimately, this will lead to more reliable identification of

**Data availability statement:** The Fastq files have been deposited in the Sequence Read Archive (SRA) (www.ncbi.nlm.nih.gov/sra) under accession number PRJNA1240204.

**Funding:** The author(s) received no specific funding for this work.

**Competing interests:** The authors have declared that no competing interests exist.

EGs and potential therapeutic targets in *P. aeruginosa*, advancing our understanding of this important pathogen.

## Author summary

Tn-Seq analyses are a crucial resource for understanding gene function and identifying potential new therapeutic targets. However, during our analyses, we encountered vast array of available tools and disparity in results reported in the literature. Therefore, we conducted a comparative evaluation of bioinformatics tools based on two gold-standard datasets of essential genes for *Pseudomonas aeruginosa* that we established. This approach enables researchers performing Tn-Seq analysis to assess the quality of their results, thereby promoting consistency and harmonization across studies.

## Introduction

Transposon Sequencing (Tn-Seq) is a high-throughput technique that utilizes transposon mutant libraries to assess the fitness or essentiality of genes under specific conditions. The use of a mariner transposon, which inserts into a thymine-adenine (TA) site only once per bacterium and in a random manner, enables the generation of a vast number of mutants [1]. Identifying essential genes (EGs) can subsequently lead to the discovery of novel therapeutic targets.

Analysis of Tn-Seq data involves several critical steps. First, data pre-processing is performed, which includes demultiplexing and adapter trimming. Then, the sequences are mapped to a reference sequence. The alignment file is then converted into a more compact format, such as the wiggle format (.wig), where the number of reads at each TA site is encoded. From this compact file, and after potential filtering steps, statistical methods are employed to determine gene essentiality. When multiple conditions are considered, additional statistical methods can be used to identify genes exhibiting significant variability in insertion counts across multiple conditions.

A wide range of algorithms has been developed for Tn-Seq data analysis. These include Hidden Markov Model (HMM)-based methods for identifying essential sites and regression analyses that utilize gene saturation or runs of consecutive empty sites. These approaches are implemented in tools such as TnseqDiff [2], ESSENTIALS [3], Magenta [4], Tnseq Explorer [5], ARTIST [6], TRANSIT2 [7] and TSAS [8]. Among them, TRANSIT2 is the most widely used, offering a comprehensive suite of tools for analyzing Tn-Seq data from pre-processing to gene enrichment. Another statistical method, FiTnEss (Finding Tn-Seq Essential genes), was introduced in a study aimed at defining the core essential genome of *P. aeruginosa* [9].

Given the variety of statistical methods, tools, and parameters available, selecting the most appropriate pipeline for reliable analysis of a specific dataset can be challenging. Some studies have evaluated specific steps of Tn-Seq analysis. For

instance, Larivière *et al.* [10] compared the lists of EGs generated using different methods (linear regression and Gumbel from TRANSIT2) with those initially retrieved and published. They proposed a ready-to-use workflow, freely accessible *via* the Galaxy platform, based on TRANSIT2 for Tn-Seq data analysis. A major limitation of available studies is the absence of a gold-standard set of EGs for evaluating the analysis process. Considering the original study as a gold-standard is suboptimal, as its results were obtained using a tool that may not be the most optimal.

In the major human pathogen *P. aeruginosa*, the loss of the porin OprD confers resistance to the last resort carbapenem antibiotics [11]. Tn-Seq analysis of the wild-type PA14 strain (PA14 WT) has identified the loss of the *oprD* gene as enhancing *in vivo* fitness in the context of gastrointestinal and spleen colonization in mice [12,13]. It is also associated with increased virulence in a mouse model of acute pneumonia [14]. However, the mechanism underlying this increased virulence of PA14 OprD-deficient (PA14 Δ*oprD*) remains unclear. Determining EGs of PA14 Δ*oprD* could help identify potential targets that provide insight into the pathways driving this increased pathogenicity. To address these questions, we first generated a highly saturated PA14 Δ*oprD* Tn insertion bank intended for comparison with PA14 WT, which required validation prior to any in-depth analysis. Therefore, we assessed the quality of our Tn-Seq analyses by sequencing a highly saturated bank of different Tn insertion mutants of PA14, in which the *oprD* gene had been cleanly deleted.

We aimed to evaluate the bioinformatics analysis of Tn-Seq data with data generated from two strains grown in Lysogenic Broth (LB): PA14 WT and PA14 Δ*oprD*. Firstly, we introduced two gold-standard datasets: a first list containing 115 genes specific to PA14 grown in LB and the second comprising 84 EGs representing a general gold-standard that can be used for any *P. aeruginosa* Tn-Seq study. Secondly, we evaluated several methods and parameter settings for Tn-Seq analysis. Our choices were guided by several criteria: the tool should (i) be free and open-source, (ii) be regularly updated, (iii) offer multiple statistical methods and adjustable parameters, and (iv) be widely used. TRANSIT2, which is a complete re-implementation of the original version in Python, fulfills all these criteria and additionally provides a graphical user interface for users who are not comfortable with command-line tools. We also included FiTnEss in our evaluation because it was specifically developed for PA Tn-Seq data [9].

## Results

### Data description

Sequencing was carried out in two separate runs: samples S1 and S2 were processed in RUN1, while samples S3, S4, S5, and S6 were processed in RUN2. This resulted in six samples corresponding to PA14 WT or PA14 Δ*oprD* grown in LB culture medium (Table 1).

### Definition of gold-standard sets of essential genes for *P. aeruginosa*

Many previous studies have reported the EGs for several *P. aeruginosa* strains grown on different culture media, using diverse biological and analytical approaches (Table 2) [9,12,15–17]. Our objective was not to generate an exhaustive catalog of EGs, but rather a reduced and reliable set that could serve as a reference for validation — a *gold standard* list

**Table 1. Description of samples used in this study.**

| RUN | Samples | Strain |
|---|---|---|
| RUN1 | S1 | PA14 Δ*oprD* |
| RUN1 | S2 | PA14 Δ*oprD* |
| RUN2 | S3 | PA14 WT |
| RUN2 | S4 | PA14 WT |
| RUN2 | S5 | PA14 Δ*oprD* |
| RUN2 | S6 | PA14 Δ*oprD* |

**Table 2. Summary of studies on essential genes in *P. aeruginosa*.**

| Dataset | Year | #EGs[a] | Strain | Media | REF | Mapper | Statistical method to identify EGs | Biological method |
|---|---|---|---|---|---|---|---|---|
| **Liberati** | 2006 | 335 | PA14 | LB | [15] | BLAST | Study of the insertion-site distribution and identification of genes that were not disrupted | Insertion sites identified by arbitrary PCR and Sanger sequencing |
| **Skurnik** | 2013 | 634 | PA14 | LB | [12] | CLC Genomics Workbench | Genes with <10 sequencing reads were considered as essential genes | INSeq pipeline |
| **Turner** | 2015 | | | | [17] | Bowtie2 | Home-made scripts inspired by ESSENTIALS package software (Monte Carlo approach) | Modified INSeq pipeline |
| PA14_init | | 434 | PA14 | BHI Agar | | | | |
| PA14_sputum | | 510 | PA14 | sputum | | | | |
| PAO1_init | | 336 (326) | PAO1 | BHI Agar | | | | |
| PAO1_succ | | 641 (621) | PAO1 | MOPS-succinate | | | | |
| **Lee** | 2015 | 352 (349) | PAO1 | 3 media[b] | [16] | MAQ v0.7.1 | Based on transposon insertions per kilobase and transposon sequence reads per kilobase | TnSeq circle method |
| **Poulsen** | 2019 | | 9 PA strains[c] | 5 media[d] | [9] | | | TnSeq |
| FWER | | 437 | PA14 | LB | | Bowtie | FiTnEss maximal stringency | |
| FDR | | 596 | PA14 | LB | | Bowtie | FiTnEss high stringency | |
| HMM | | 316 | PA14 | LB | | BWA | Transit HMM | |
| CORE | | 321 | Core pangenome (9 PA strains) | 5 media | | Bowtie | FiTnEss high stringency | |

[a]EGs gives the number of EGs identified in each dataset. The numbers in parentheses indicate the number of orthologous EGs from the PAO1 strain found in the PA14 strain.

[b]LB, MOPS-pyruvate, and media made from cystic fibrosis sputa [16].

[c]Clinical trains representing major clusters of the whole genome analysis of 2,560 *P. aeruginosa* genomes in the National Center for Bio-technology Information Genome database [9].

[d]Fetal bovine serum, Human serum, LB, Minimal M9, Synthetic cystic fibrosis sputum Media and urine [9].

of genes that should be consistently identified as essential across all Tn-Seq experiments. To construct this gold standard dataset, we postulated that genes identified as essential in multiple independent studies, each employing distinct biological systems, mapping strategies, and statistical methods, can be considered high-confidence EGs. Technically, intersecting EG lists from several studies allowed us to extract genes that were consistently identified as essential across different experimental and analytical frameworks. Initially, we attempted to provide the minimal set of EGs of PA14 WT grown in LB. This will serve as the gold-standard EGs expected to be retrieved in any EGs set for PA14 under standard condition. To obtain this gold-standard list, we considered only previous datasets dealing with PA14 grown in LB, including Liberati, Skurnik, Poulsen_FWER, Poulsen FDR and Poulsen_HMM datasets (Table 2), but we took only into account the Poulsen_FWER dataset as FWER genes are included in the FDR dataset. Intersecting these four datasets (Fig 1A), 115 EGs were identified as common across all four studies and could be considered as forming the gold-standard list of EGs for PA14 grown in LB (GOLD_115 which is therefore strain and condition specific and may include genes whose essentiality is conditional on the PA14 background or the LB medium). Next, we aimed to obtain a more generic EG list for any *P. aeruginosa* strain in any culture medium. Poulsen *et al.* [9] introduced the core EGs for several strains of PA, combining nine strains across five media which resulted in a list of 321 EGs (Table 2). By intersecting the Poulsen_CORE

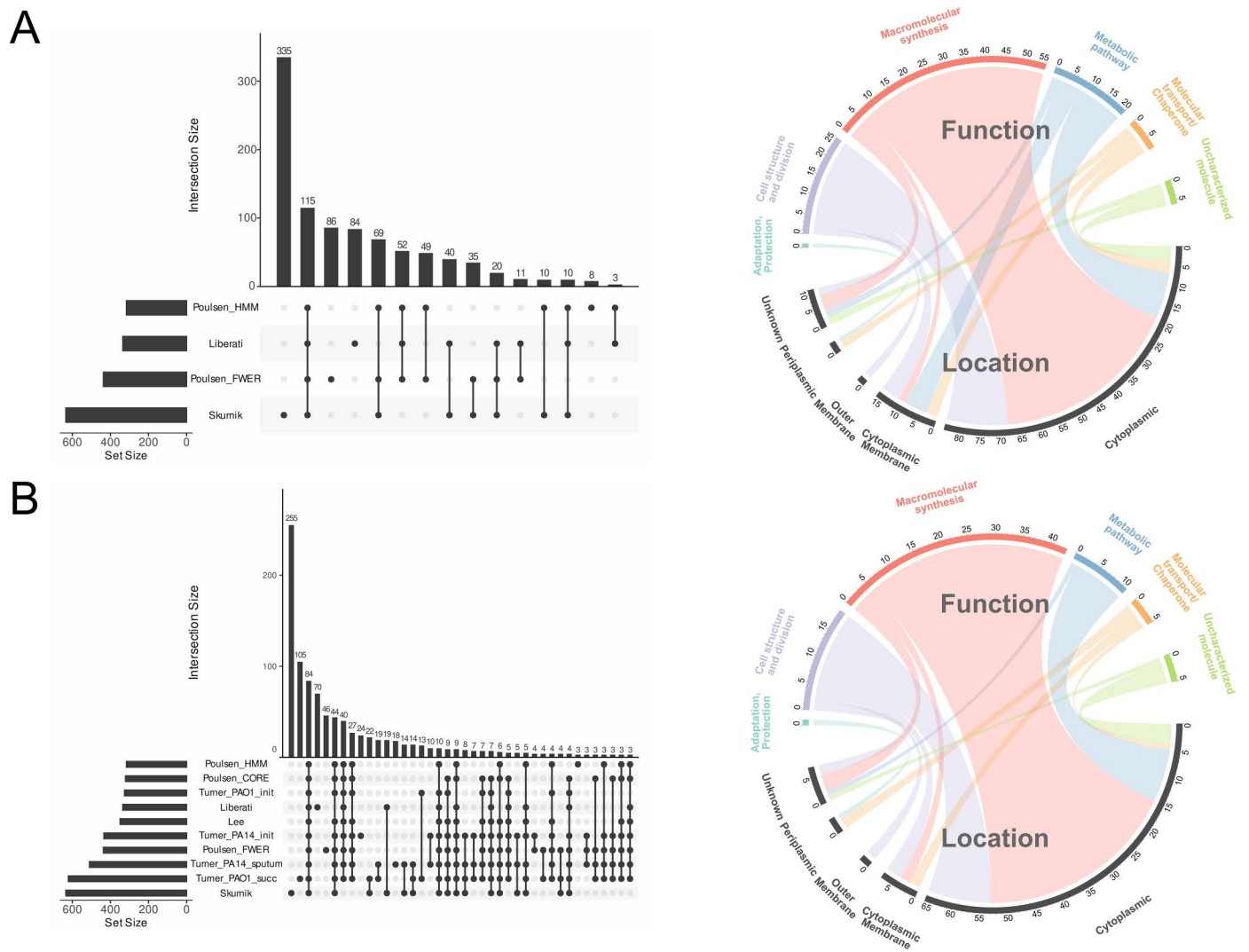

**Fig 1. Determination and characterization of the gold-standard essential genes. (A)** Upset plot displaying the intersections of EGs for PA14 grown in LB and chord diagram of the GOLD_115 essential genes list. **(B)** Upset plot displaying the intersections of EGs for several strains of PA grown in different culture media and chord diagram of the GOLD_84 essential genes list. In the upset plots, each vertical bar represents a distinct set of genes, and the plot highlights the overlapping gene sets that are considered essential across the studies. The chord diagrams represent the relationship between subcellular location (bottom) and general function (top), with the number of genes for each category indicated.

dataset with the Poulsen_FWER and Poulsen_HMM datasets, as well as four datasets from Turner *(*PA14_init*,* PA14_sputum*,* PAO1_init*, and* PAO1_succ*),* and the datasets from Liberati, Skurnik, and Lee (Fig 1B), we obtained a reduced but reliable list of 84 EGs, which can be used for any Tn-Seq analysis of PA (GOLD_84, representing a more general "core" set of essential genes conserved across strains and growth conditions). Finally, these ten datasets encompassed PA14, PAO1 and the core genomes of 9 PA strains, including PA14 and 8 clinical strains, grown in several media (Brain Heart Infusion (BHI), Fetal Bovine Serum, Human serum, LB, Minimal M9, MOPS-pyruvate, Synthetic Cystic Fibrosis Sputum Media, Urine, MOPS-Succinate). The GOLD_115 and GOLD_84 could evolve dynamically with contributions from future studies. Adding new data from new PA strains and media would inevitably reduce our list and ultimately refine it. The two gold-standard EG lists are available in S2 Table. GOLD_115, genes are mainly involved in macromolecular synthesis

and metabolism (47.93%), cell structure and division (21.74%), and metabolic pathways (17.39%). A similar distribution is observed in GOLD_84, with 51.19%, 22.62%, and 13.1% of genes in the respective categories. Most genes are predicted to be cytoplasmic, with 72.17% in GOLD_115 and 77.38% in GOLD_84. Thus, while the overall functional profiles of the two sets are similar, GOLD_115 reflects a narrower, condition-specific set, whereas GOLD_84 represents a broadly conserved core of essential genes.

**Assessment of Tn-Seq analysis methods.** To determine which analysis method would provide the most reliable set of EGs from our Tn-Seq data, we compared the finely tunable TRANSIT2 suite with different statistical methods and parameters, using both bowtie and BWA mappers as well as FiTnEss for generating sets of EGs. Next, we compared the resulting datasets with our GOLD_115 and GOLD_84 lists to determine whether they included, at a minimum, the gold-standard EGs.

**Impact of the mapper on identifying essential genes.** After pre-processing steps (adapters trimming, read filtering, etc. [5,7]), a major step is to accurately map sequences against the reference genome, as this mapping is the base to count the number of reads at each TA site. Previous studies [9,10] have shown that the choice of the mapper impacts the results: bowtie [18] is better suited for short sequences generated in Tn-Seq experiments, whereas BWA [19] is commonly used in Tn-Seq analysis and is included in the TPP package of TRANSIT2 for data pre-processing.

Firstly, we compared the quality metrics of the transposon libraries obtained with the *tnseq_stats* function of TRANSIT2 from both bowtie and BWA mappings (see Material and methods for more details about parameters). We observed that the quality metrics obtained with BWA were significantly different, falling below the recommended thresholds, when compared to those observed with bowtie (Table 3). Although, there are no formal criteria for determining what constitutes a good library and for excluding certain samples [7], one important metric is density, defined as the number of TA site with at least one insertion divided by the total number of TA sites, which should be above 30% and ideally above 50% [20]. In our study, density was above 50% for all the samples whatever the mapper. The percentage of mapped reads was higher with BWA. NZ-mean values, corresponding to mean over non-zero sites, which should be above 10 and ideally above 50, were above 50 for all samples with both mappers, but were below 20 for sample S6. The max_ct values, corresponding to the highest counts across all TA sites to check for outlier, were 14–70 times higher with BWA. Skewness, a measure of the read count distribution, indicate sample noise when it exceeds 50. The skewness values were low, ranging from 4.1 to 14.7 with bowtie, but were greater than 50 with BWA, ranging from 194.8 to 248. The two other measures of read count distribution, Kurtosis and Pickands tail index, confirmed this trend. These results showed that the quality metrics depends on the mapper used, with bowtie appearing to yield better results.

**Table 3. Quality metrics obtained with the *tnseq_stats* function of TRANSIT2 from both bowtie and BWA mappings.**

|  | S1 | | S2 | | S3 | | S4 | | S5 | | S6 | |
|---|---|---|---|---|---|---|---|---|---|---|---|---|
|  | bowtie | BWA | bowtie | BWA | bowtie | BWA | bowtie | BWA | bowtie | BWA | bowtie | BWA |
| **Number of reads** | 8003192 | 8003192 | 8713300 | 8713300 | 6806023 | 6806023 | 10967540 | 10967540 | 9955879 | 9955879 | 1148038 | 1148038 |
| **Mapped_read** | 6714836 | 7573812 | 7269135 | 8101194 | 5059919 | 5927396 | 8860617 | 10068873 | 7197141 | 8444659 | 786779 | 938661 |
| **%mapped_reads** | 83.90 | 94.64 | 83.43 | 92.98 | 74.34 | 87.09 | 80.9 | 91.81 | 72.29 | 84.83 | 68.53 | 81.77 |
| **Density** | 0.599 | 0.661 | 0.661 | 0.720 | 0.643 | 0.693 | 0.659 | 0.719 | 0.674 | 0.730 | 0.539 | 0.571 |
| **mean_CT** | 66.4 | 73.5 | 71.9 | 78.5 | 50.0 | 55.9 | 87.6 | 96.3 | 71.2 | 78.4 | 7.8 | 8.7 |
| **NZ_mean** | 110.9 | 111.2 | 108.8 | 109.0 | 77.8 | 80.8 | 132.9 | 133.9 | 105.7 | 107.5 | 14.4 | 15.3 |
| **NZmedian** | 56 | 53 | 51 | 48 | 35 | 34 | 55 | 52 | 47 | 45 | 6 | 6 |
| **Max_ct** | 3284 | 235554 | 3942 | 160445 | 10661 | 185717 | 17109 | 251728 | 4579 | 115350 | 964 | 20780 |
| **Total_cts** | 6704929 | 7420085 | 7259889 | 7920051 | 5049342 | 5646221 | 8841568 | 9717900 | 7187530 | 7917388 | 785172 | 880423 |
| **Skewness** | 4.1 | 247 | 4.4 | 240.5 | 14.7 | 248.0 | 11.9 | 246.4 | 4.9 | 194.8 | 6.6 | 200.7 |
| **Kurtosis** | 30.4 | 62832.6 | 36.9 | 62362.3 | 857.8 | 64129 | 590 | 64377 | 46.6 | 46661.4 | 98.8 | 45162.2 |
| **Pickands_tail_index** | 0.155 | 0.151 | 0.133 | 0.185 | 0.182 | 0.631 | 0.237 | 0.538 | 0.245 | 0.542 | 0.271 | 0.606 |

Interestingly, we noted that the percentage of mapped reads was lower for sample S6 compared to the other samples with around 68% and 81% of reads mapped with bowtie and BWA, respectively. In addition, sample S6 showed NZ-mean values, which ideally should be greater than 50, equal to 14.4 with bowtie and 15.3 with BWA. Computing the correlation matrix (Fig 2), we confirmed that sample S6 was an outlier. The correlation coefficients were higher between samples of the same strains (S3 and S4 for PA14 WT, and S1, S2, S5 and S6 for PA14 ΔoprD). Regarding replicates, the coefficients were lower with sample S6. Finally, we discarded sample S6, and used samples S1, S2 and S5 for PA14 ΔoprD and samples S3 and S4 for PA14 WT.

Secondly, we compared the lists of EGs obtained from both mappers with several statistical methods available in TRANSIT2 (Gumbel, HMM and TTN-Fitness) for PA14 WT (S3 and S4) and PA14 ΔoprD (S1, S2 and S5). The Gumbel method determines the probability of gene essentiality using a Bayesian model based on the longest consecutive sequence of TA sites without insertion. Note that we considered as essential both genes classified with the Gumbel and the binomial models. The HMM method, based on Hidden Markov Model, assesses the essentiality of entire genomes and can identify growth advantage (GA) and growth defect regions (GD) in addition to the essential and non-essential categories. The HMM function also provides additional columns with confidence information classified in two categories: the

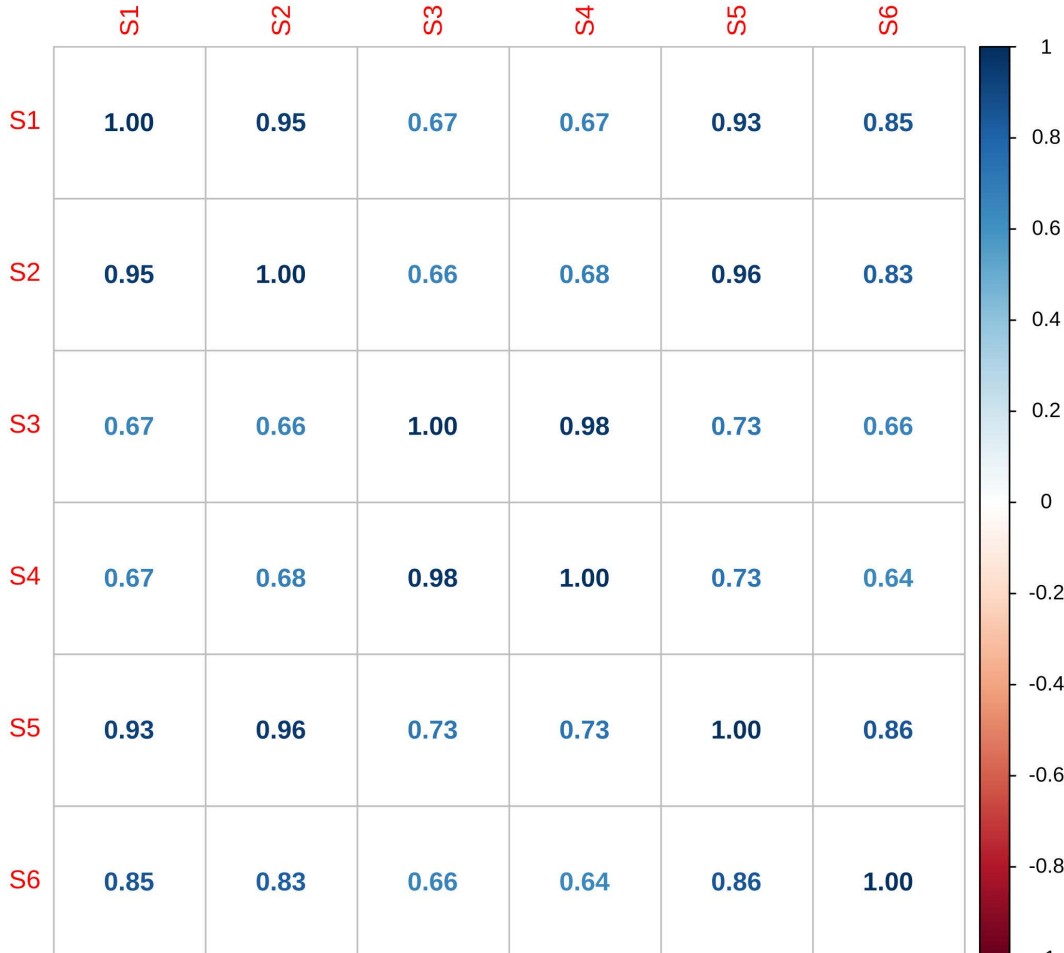

**Fig 2. Correlation matrix between samples.** Numbers correspond to Pearson correlation coefficients obtained using the corrplot R package from the mean counts for each gene in each sample generated with the *gene_means* function of TRANSIT2.

*low-confidence category,* meaning the probability of the HMM call is low, so the HMM call should be ignored and the *ambiguous category,* meaning there is another state with higher probability; these cases are borderline, where the gene could fall into either category. Finally, the TTN-Fitness method estimates gene fitness in a single condition, while correcting for biases in Himar1 insertion preferences at TA sites based on surrounding nucleotides. Notably, this method uses the Gumbel results as input to classify genes into GD and GA categories, in addition to essential and non-essential ones. Fig 3A shows that BWA consistently returned a smaller number of EGs compared to bowtie. In most cases, BWA produced lists with very few EGs, fewer than 50 genes using the Gumbel and HMM methods, except when the HMM method considered both essential and GD categories, yielding 462 and 479 genes for PA14 WT and PA14 Δ*oprD* strains, respectively. Regarding the HMM method, the proportions of low-confidence and ambiguous genes were higher with BWA compared to bowtie. Unexpected results were observed with TTN-fitness, while BWA yielded stable results, bowtie identified an exceptionally high number of EGs, exceeding 5,900 genes.

Thirdly, we examined the presence of the gold-standard genes in the lists of EGs obtained from both mappers. Fig 3B shows that for both GOLD_115 and GOLD_84, fewer than 10% of genes were retrieved with BWA using Gumbel, TTN-fitness and HMM methods whereas bowtie retrieved 37%, 82% and 54% for PA14 WT. For PA14 Δ*oprD*, the percentage of retrieved GOLD_84 genes was lower. No gene from the gold standard dataset were identified using BWA with Gumbel, TTN-fitness and HMM, but 32%, 80% and 11% of GOLD_84 were identified respectively with bowties. Interestingly, when considering both essential and GD genes, the HMM method identified more than 90% of gold-standard EGs (GOLD_84 and GOLD_115) for both mappers (Fig 3B) and both strains. Only around 80% of gold-standard genes were retrieved within the TTN-fitness results obtained with bowtie despite the EG list containing nearly 6,000 genes. We thereby excluded TTN-fitness of the remainder of our analysis.

Altogether, these findings revealed that the bowtie mapper outperformed BWA allowing for the greater identification of EGs and gold-standard genes. Therefore, only the results obtained with the bowtie mapper are presented in the following sections of our study.

We also studied the impact of parameters on identifying EGs (see S1 Text and S1 Table) and showed that the default replicate handling options, *Sum* for Gumbel and *Mean* for HMM, yielded slightly better results. Additionally, enabling the LOESS option in HMM provided a minor improvement. Therefore, the default parameters for replicate handling and the LOESS option were used for the remainder of the study. An important step in Tn-Seq analysis is the normalization of read counts as it ensures that other sources of variability are not mistakenly interpreted as real differences in datasets. We then studied the impact of normalization on identifying EGs (see S2 Text and S1–S5 Figs) by evaluating the seven normalization methods offered in TRANSIT2. The *quantile* normalization showed the best results by using only genes flagged as essential in the HMM model. However, considering essential and GD genes, the choice of the normalization method had less impact and led to more stable gene lists, except for *betageom* and *zinfnb* normalizations, which should be avoided.

### Impact of statistical method on providing sets of essential genes

In this part, we studied the impact of the main statistical methods in EG identification using bowtie. The Gumbel (used with default parameters) and HMM (used with optimized parameters, i.e., quantile normalization and LOESS option) methods from TRANSIT2 were evaluated in addition to FiTnEss [9]. The important features of FiTnEss are that (i) it evaluates genes (rather than individual TA sites or stretches of TA sites) and (ii) it uses a simple two-parameter model to capture all the features of the data. To vary the stringency, two different levels of multiple testing adjustments can be applied: one with maximal stringency, which provides the most reliable set of essential genes [family-wise error rate (FWER)], to identify genes with few or no sequencing reads, and the other with high stringency but slightly relaxed [false discovery rate (FDR)] to identify genes that are statistically significant but contain a low number of reads.

Table 4 shows the number of EGs identified using the Gumbel, HMM and FiTnEss methods. Approximately 10–15% of genes are expected to be essential in bacteria without strong selection [7]. In the case of *P. aeruginosa* between 610

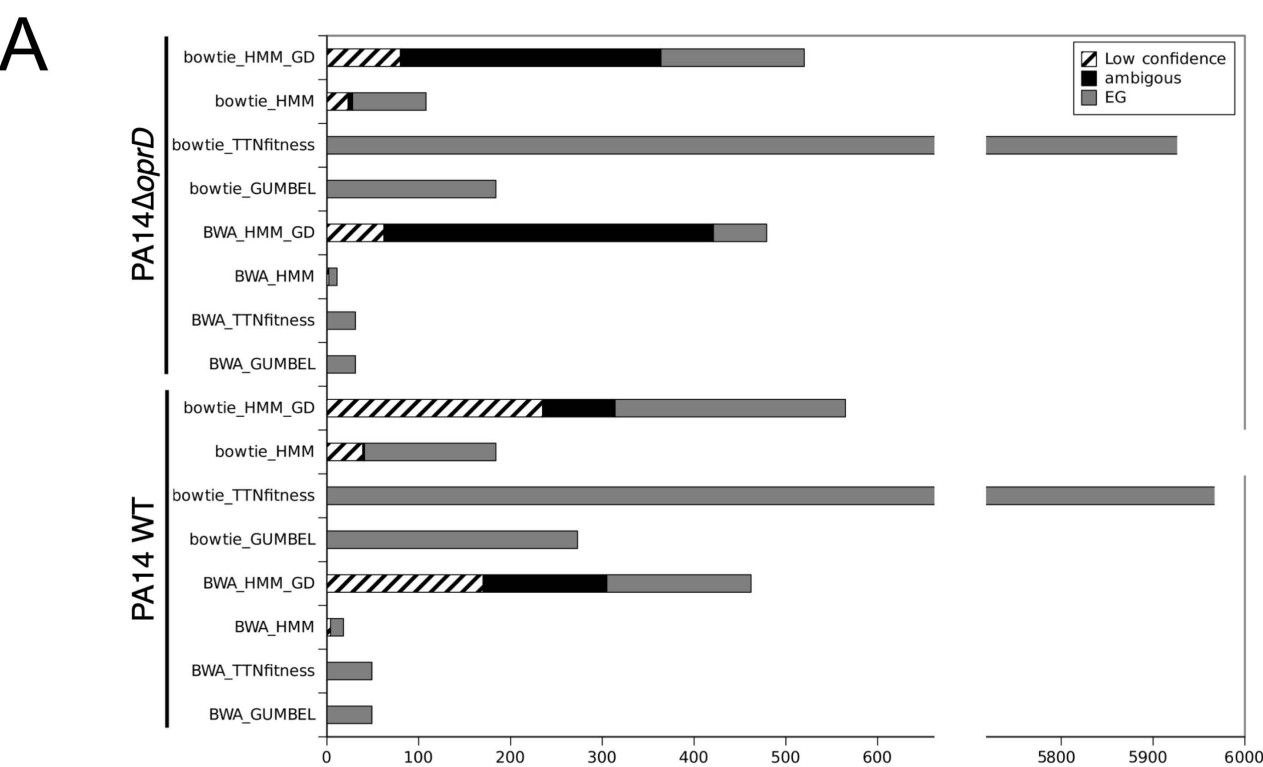

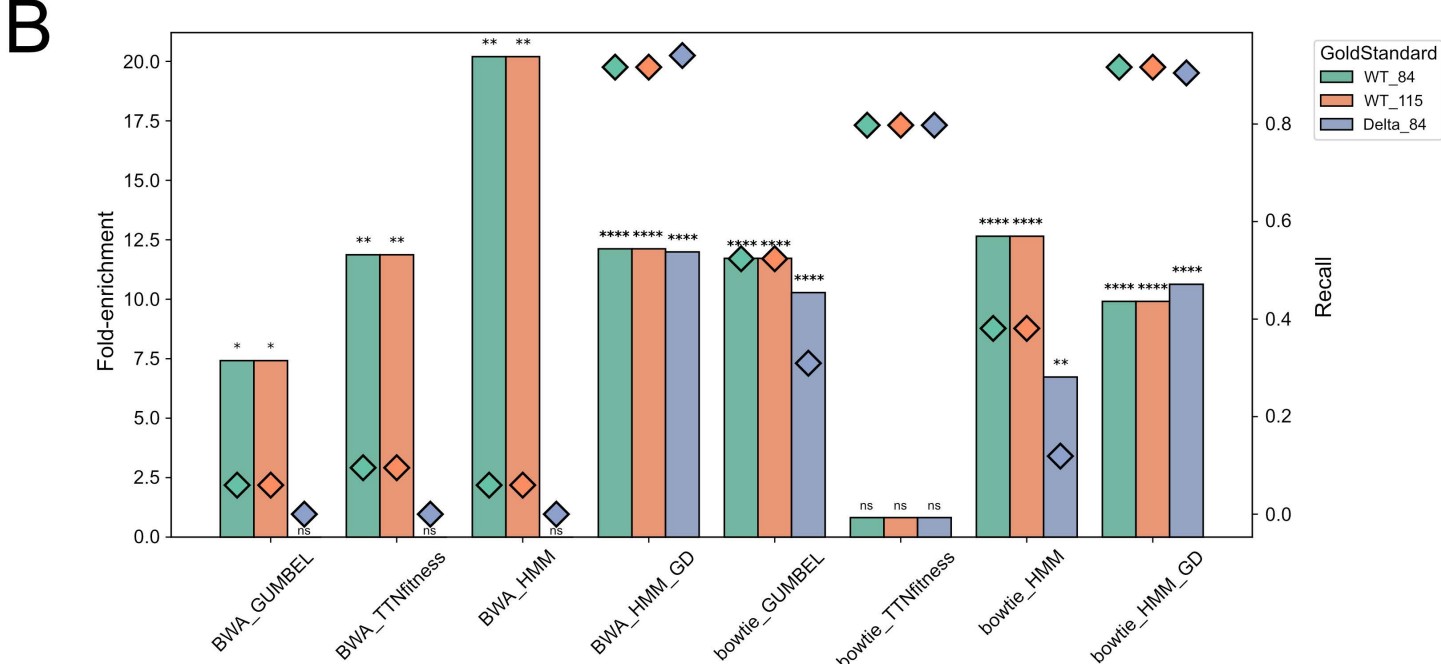

**Fig 3. Impact of bowtie and BWA mappers on the identification of essential genes. (A)** Number of EGs identified with Gumbel, TTN-fitness and HMM methods, considering either only the essential category (HMM) or both the essential and Growth Defect categories (HMM_GD). For the HMM method, the proportions of low-confidence and ambiguous classifications are also displayed. **(B)** Bar plots show fold enrichments with associated *p*-values, and diamonds indicate the recall of gold-standard genes identified by each method. Colors denote gene sets: green, 84 core EGs for PA14 WT; orange, 115 gold-standard genes for PA14 WT; and purple, 84 core EGs for PA14 Δ*oprD*.

**Table 4. Impact of statistical method on identifying essential genes.**

| | EG[a] | | GOLD_84 | | GOLD_115 |
|---|---|---|---|---|---|
| | **PA14 WT** | **PA14 ΔoprD** | **PA14 WT** | **PA14 ΔoprD** | **PA14 WT** |
| **GUMBEL** | 273 | 184 | 44 | 26 | 62 |
| **HMM** | 346 | 346 | 65 | 60 | 87 |
| **HMM_GD** | 651 | 640 | 78 | 79 | 105 |
| **FiTnEss_FWER** | 358 | 269 | 80 | 67 | 103 |
| **FiTnEss_FDR** | 609 | 510 | **84** | **84** | **115** |

[a]The "EG" column gives the number of essential genes returned by each method for the WT and PA14 ΔoprD conditions. The columns "GOLD_84" and "GOLD_115" represent the number of EGs intersecting with the respective gold-standard datasets. Note that the GOLD_115 dataset corresponding to PA14 WT was not applied to PA14 ΔoprD. The best values appear in bold.

and 920 EGs are expected. This range was obtained with HMM_GD, considering essential and GD categories, and with FiTnEss FDR. As expected and shown before, the Gumbel method identified very few EGs, 273 for PA14 WT and 184 for PA14 ΔoprD. Around 50% of gold-standard genes were present in the WT gene sets, but only 30% of GOLD_84 for PA14 ΔoprD. Contrary to the HMM method, which determines essentiality at the genome-wide scale and classifies genes into essential, growth-defect, growth-advantage, and non-essential categories, the Gumbel method performs a gene-by-gene analysis of insertions at TA sites and distinguishes only between essential and non-essential genes. Working with only two categories at the gene level likely explains the lower number of essential genes returned by the Gumbel method. HMM_GD produced better results, identifying more than 90% of gold-standard genes identified in the list obtained when considering essential and GD genes for both strains. The stringent FiTnEss method (FWER) produced a list of 358 and 269 EGs for PA14 WT and PA14 ΔoprD, respectively, containing around 95% and 80% of GOLD_84 and 90% of GOLD_115 for PA14 WT. Using the relaxed FDR method led to an EGs list of 609 genes containing 100% of the gold-standard genes (GOLD_84 and GOLD_115) for PA14 WT and 510 EGs containing 100% of the GOLD_84 dataset for PA14ΔoprD. FiTnEss was the only methods that successfully identified all the gold-standars genes. It consistently returned a reliable list of EGs. Finally, by comparing the lists of EGs identified by the different statistical methods, we found that a substantial number of EGs were shared across methods. For PA14 WT, 297 genes were common in both HMM_GD and FiTnEss_FDR and 183 were shared between HMM_GD, FiTnEss_FDR and Gumbel. In contrast, only around one hundred genes were uniquely identified by either HMM_GD or FiTnEss_FDR (S6 Fig). A similar pattern was observed for PA14Δ oprD (S7 Fig).

Our results indicated that the HMM_GD method, when considering both essential and GD genes, and the FiTnEss approach, when using FDR, are the most reliable strategies for Tn-Seq analysis in *P. aeruginosa*.

## Methods to determine conditional gene essentiality

Several methods are available to determine which genes exhibit statistically significant variability of insertion counts across multiple conditions. We evaluated four statistical methods available in TRANSIT2: resampling (based on a permutation test) and Mann-Whitney U-test (utest) for pairwise comparisons; ANOVA and ZINB, allowing to compare multiple conditions. In addition, we also used the results of FiTnEss and HMM to determine conditional gene essentiality. It is noteworthy that FiTnEss and HMM provide EG lists for individual conditions (here, either PA14 WT or PA14 ΔoprD) while the statistical methods, which account for the raw read counts that exhibit significantly different read counts between the conditions but are not necessarily classified as EGs. LB. Therefore, for FiTnEss and HMM, the EG lists obtained for PA14 WT and PA14 ΔoprD were intersected to identify the essential genes specific to PA14 WT and PA14 ΔoprD grown in LB. Genes not present in the intersection of the PA14 WT and PA14 ΔoprD EGs lists were considered conditionally essential.

Firstly, we looked for the number of conditional essential genes we could retrieve using these methods. Among the statistical methods, resampling provided a much higher number of EGs (1,235) than utest (18) ANOVA (22) and ZINB (247)

(Fig 4A). Intersecting the EG lists retrieved with FiTnEss_FDR and HMM_GD found a number of conditional genes similar to ZINB (157 with HMM_GD and 197 with FiTnEss_FDR). As the intersection may lead to false positives (*i.e.,* genes that are not identified as essential in the other strain because they are missing in only one sample), we filtered out the genes specific to one strain that were not classified as non-essential in all samples in the other strain. This stringent filter did not impact the list produced with HMM_GD, as the HMM script processes all replicates simultaneously. In contrast, the filter reduced the number of genes for FiTnEss from 197 to 117 genes (Fig 4A). The number of genes specific to PA14 WT, *i.e.,* genes showing fewer read counts than for PA14 ΔoprD or classified as essential in PA14 WT but not in PA14 ΔoprD, was higher with the intersection methods (HMM and FiTnEss) than in the statistical methods, where it was more balanced. Indeed, with fewer replicates such as PA14 WT, which contains two replicates, the intersection may lead to more false positives. Then we compared the list of genes with an UpSet plot, discarding the resampling method which returned a very high number of genes compared to all other methods, and FiTnEss FDR_filter as it is included in FiTnEss_FDR. As shown in Fig 4B very few genes were common between the methods. The intersecting methods (HMM_GD and FiTnEss_FDR) shared 31 genes whereas ZINB and FiTnEss_FDR shared 17 genes. Twelve genes were common in ZINB, HMM_GD and FiTnEss_FDR. Only one gene was retrieved by all methods, PA14_RS20600 (PA14_50730), which encodes *shaF*. This latter is part of a six-gene cluster (PA14_50680–PA14_50730) encoding the Sha system, a sodium/proton (Na+/H+) antiporter complex. This system plays a pivotal role in monovalent ion homeostasis, particularly by mediating sodium extrusion, thereby conferring sodium tolerance and contributing to the bacterium's virulence. Specifically, *shaF* encodes one of the protein components of this multi-subunit transporter. Because well-established positive controls for conditional essentiality are scarce, we sought to experimentally validate *shaF* that was consistently identified as conditionally essential across all methods. Clean deletion mutants were attempted in both genetic backgrounds (S3 Table). While the *shaF* deletion was successfully obtained in the PA14 WT strain, repeated attempts to delete *shaF* in the PA14 ΔoprD background were unsuccessful, indicating that *shaF* is specifically essential in the PA14 ΔoprD strain.

Secondly, as the GOLD_84 dataset consists in gold-standard genes of *P. aeruginosa*, we did not expect to find these EGs in our conditional gene essentiality list, as they would be essential in both conditions. No GOLD_84 gene was found, except for one gene with the resampling method and three genes with HMM_GD. Eventually, we verified that all methods reported the *oprD* gene (deleted in PA14 ΔoprD) as essential in PA14 ΔoprD while not in PA14 WT, which was the case for all methods except for utest which did not report this gene.

Altogether, these findings showed that each method to determine conditional gene essentiality returned its own list of EGs. Among the statistical methods, the resampling approach produced a large number of conditional essential genes compared to the other methods and should therefore be avoided. The ANOVA and utest methods yielded very few genes, and moreover, utest did not report the *oprD* gene as conditional essential in PA14 ΔoprD. Consequently, these two methods should also be avoided. Finally, the ZINB method appears to yield consistent results and can be used to determine conditional gene essentiality. Regarding the intersection-based methods, as shown previously, FiTnEss_FDR provided more robust EG identification. Here, we showed that no GOLD_84 gene was identified as conditionally essential with FiTnEss_FDR, whereas three GOLD_84 genes were detected with HMM_GD; therefore, FiTnEss_FDR should be preferred. Accordingly, we propose that FiTnEss_FDR complemented with information retrieved from ZINB, would be an accurate and reliable method to determine conditional essential genes in PA. Indeed, ZINB, like the other statistical methods, relies on raw read counts and identifies genes showing significantly different read counts between the two conditions; however, these genes are not necessarily classified as essential. In contrast, FiTnEss_FDR considers only essential genes that are specific to one condition. Therefore, if the user aims to identify strictly conditional essential genes, we recommend using FiTnEss_FDR, and using ZINB otherwise. ZINB outputs an informative table with several values, such as the mean read count for each condition, log2 fold-change values, or gene status indicators (e.g., low numbers of TA sites in a gene). Even if the user uses the FiTnEss_FDR gene list, all the ZINB values can still help assess whether a gene can be considered conditionally essential.

PLOS Computational Biology

A

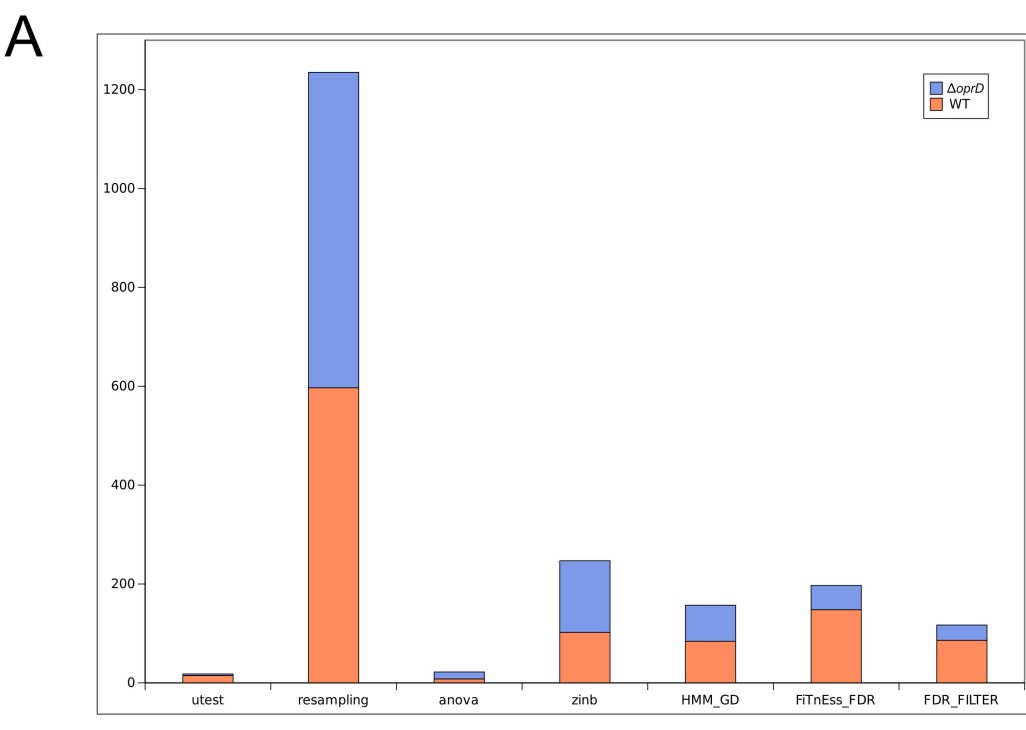

B

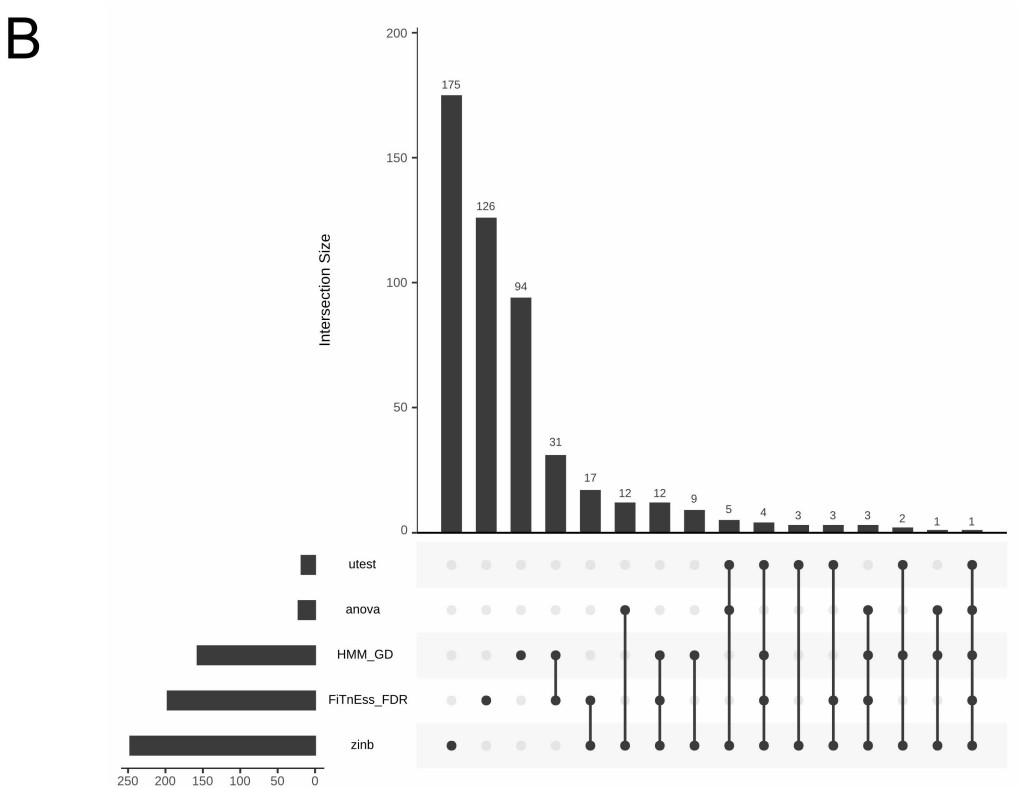

**Fig 4. Impact of the methods to determine conditional gene essentiality. (A)** Number of conditional genes depending on the method. Gene with lower read counts in WT appears in black, while those in the PA14Δ*oprD* condition appear in grey. **(B)** UpSet plot displaying the intersections of conditional genes identified by statistical methods (utest, resampling, anova, ZINB) and by intersection methods (HMM_GD and FiTnEss_FDR). Each vertical bar represents a distinct set of genes, and the plot highlights the overlapping gene sets that are considered conditionally essential across the methods.

## Discussion

A multitude of algorithms have been developed to analyze Tn-Seq data, including HMM-based methods for identifying essential sites and regression analyses using gene saturation or a series of consecutive empty sites. These approaches have been implemented in tools such as TnseqDiff [2], ESSENTIALS [3], Magenta [4], Tn-seq Explorer [5], ARTIST [6], TRANSIT [7], TSAS [8]. FiTnEss was introduced to define the core essential genome of PA, serving as another statistical method developed to identify EGs [9]. Among these tools, TRANSIT2 is the most popular, offering a wide array of functionalities for analyzing Tn-Seq data, from pre-processing to gene enrichment.

One of the primary limitations of existing studies is the absence of a gold-standard set of EGs to evaluate the analysis process. Indeed, many studies have determined EGs for *P. aeruginosa* based on their own references rather than a standardized reference gene set that could be used to initially validate any method. In this study, we attempted to address this critical gap for PA by establishing a set of gold-standard EGs (GOLD_84) for evaluating the Tn-Seq analysis process. We also introduced a second gold-standard gene set (GOLD_115) dedicated to PA14 gown in LB. Using our established gold-standard datasets will improve analysis reliability because they help in quantifying the accuracy of Tn-Seq analyses. We demonstrated that FiTnEss_FDR identified up to 100% of the gold-standard EGs, while HMM, considering both essential and GD genes, identified around 90% for both PA14 WT and PA14 Δ*oprD*. The GOLD_115 and GOLD_84 lists represent robust starting points for defining EGs in *P. aeruginosa*. However, these datasets could evolve dynamically with contributions from future studies. Open-data initiatives enabling the integration of Tn-Seq results from additional *P. aeruginosa* strains and culture media would refine our core EGs lists further. This iterative approach mirrors the methodology of Poulsen *et al.* but scales it to leverage community-wide collaboration, ensuring the EGs framework remains both precise and adaptable. Therefore, we anticipate that making our gold-standard datasets publicly available will assist the scientific community in future Tn-Seq analyses. As the GOLD_84 dataset can be applied to any PA strain and culture medium, this dataset includes information relative to PAO1 locus tag, to readily use this resource (S2 Table).

This study compared various analysis methods, tools, and parameter optimization, revealing significant differences in performance. Previous studies have shown that the choice of the mapper impacts the results [9,10]. Bowtie has been described as better suited to short sequences generated in Tn-Seq experiments compared to BWA, which is the mapper used in the TPP package of TRANSIT2 [18,19]. Here, we showed that BWA mapper performed poorly compared to bowtie, identifying less than 10% *vs.* 40–50% of gold-standard EGs, respectively. In addition, we demonstrated how different parameters of TRANSIT2 affect the results. For example, using the LOESS option in the HMM model slightly improved results. This study follows an overall workflow that emphasizes the use of gold-standard datasets to compare different Tn-Seq analysis methods (Fig 5).This extensive evaluation provides a framework for other researchers to select the most appropriate tools and parameter settings for their own Tn-Seq analyses, ensuring robust and reliable identification of essential genes.

We anticipate that this information will help in choosing the most appropriate tools for analyses and facilitate parameter optimization to improve the sensitivity and specificity of Tn-Seq analyses.

## Conclusion

We introduced reliable gold-standard datasets of EGs for PA to facilitate the comparison of different methods for Tn-Seq analysis. The GOLD_84 and GOLD_115 datasets provide a standardized reference for PA enabling the assessment of new Tn-Seq data quality and the benchmarking of bioinformatic pipelines, and make informed choices about analysis strategies tailored to their experimental conditions. The GOLD_84 dataset will allow standardization of comparison of results across studies, promoting consistency and better reproducibility of results in the field. We anticipate our results being a starting point for guiding tool development since our gold-standard datasets can serve as benchmarks for developing and improving Tn-Seq analysis tools, potentially leading to furthermore accurate, and efficient methods. By harmonizing methods of EGs identification, our work supports the discovery of potential new therapeutic targets against the major pathogen *P. aeruginosa*.

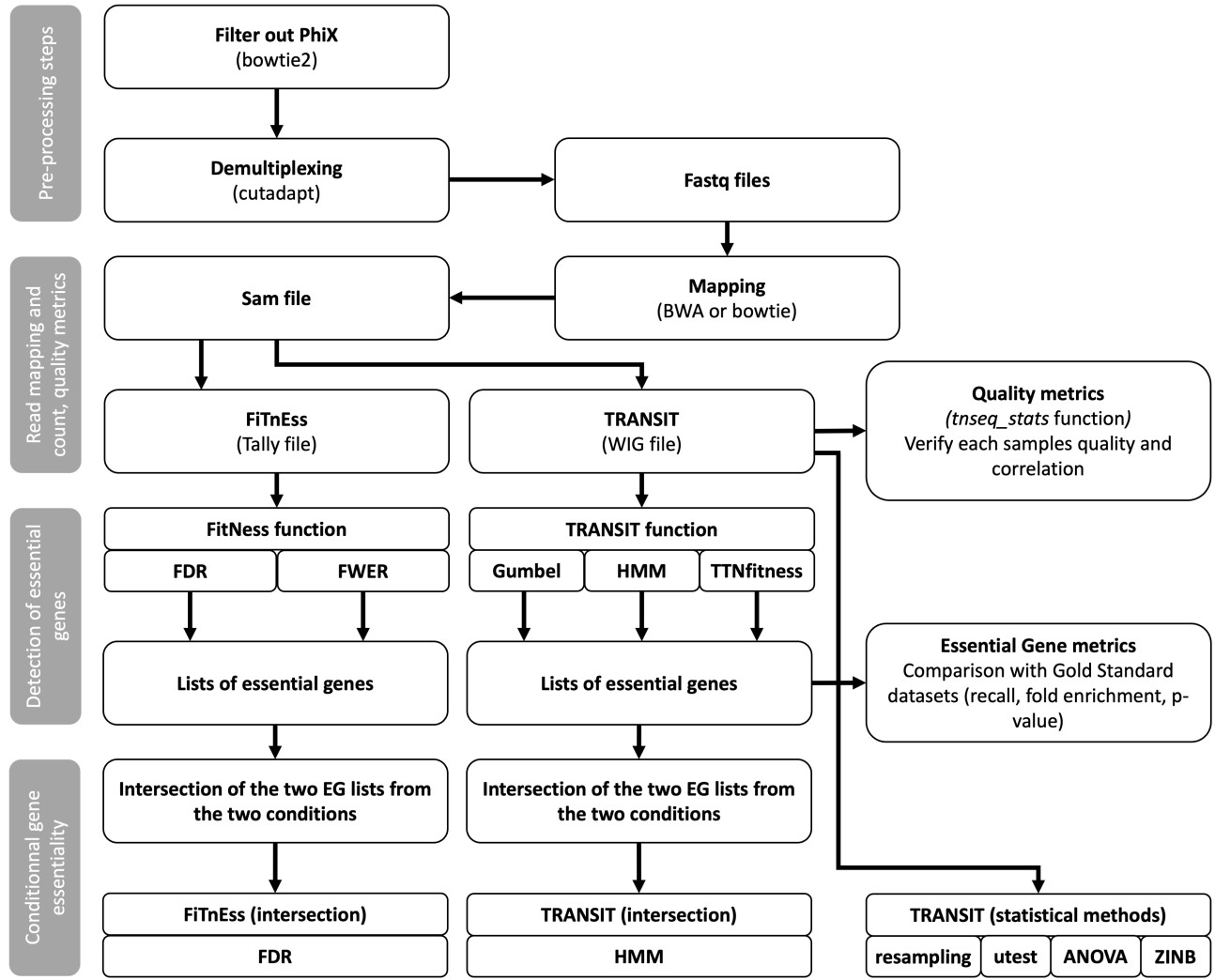

**Fig 5. Overview of the methodological workflow used in this study for the comparative evaluation of Tn-Seq analysis methods in *Pseudomonas aeruginosa*.**

## Materials and methods

### List of abbreviations

All the abbreviations used throughout the text are listed in the S4 Table.

### Bacterial strains and growth conditions

The laboratory strain *P. aeruginosa* PA14 UCBPP [21] (and its derivatives were used in this study. The PA14 parent wild-type (WT), mutant lacking OprD (ΔoprD by clean deletion) and the OprD mutant complemented with *oprD* under the control of its native promoter strains were gifts from D. Skurnik [12,13]. Bacterial cultures were incubated at 37 °C on Lysogeny Broth (LB) (BD Difco) under agitation at 250 rpm.

## PA14 Δ*oprD* TnBank preparation

To construct the TnBank, PA14 Δ*oprD* was grown overnight in lysogeny broth (LB) and *E. coli* SM10λpir/pSAM_DGH was grown overnight in LB+gentamicin 15µg/mL at 37°C at 250 rpm. Equal volumes of both cultures were mixed and centrifuged for 5 min at 3000 rpm. Pellets were resuspended with 200 µL of water for two washes, and finally resuspended in 200 µL of LB. Mattings for conjugation were performed by plating mixed bacteria in spots onto LB agar for 3 hours at 37°C to transfer pSAM_DGH into PA14 Δ*oprD*. The transposon embedded in the plasmid was randomly inserted into a single TA site per PA14 Δ*oprD* bacterium, inactivating the gene wherever it was found. The spots were then resuspended in LB and mutants selected in LB+gentamicin 30 µg/mL+irgasan 25 µg/mL at 37°C for 24–48 hours. All colonies (around 300,000 mutants) were scrapped and conserved in cryotubes, which were finally pooled in 60 mL of LB with gentamicin 20 µg/mL and cultured for 2 hours at 37°C at 250rpm. The resulting PA14 Δ*oprD* Tn Bank was dispensed into 1mL cryotubes containing glycerol for storage at -80°C.

## Tn-Seq samples preparation

TnBanks samples were prepared as described previously [12–14]. Briefly, TnBanks were grown overnight in 5mL of LB at 37°C at 250 rpm. Cultures were diluted to $OD_{600} = 1$ and centrifuged 15 min at 3000 rpm. Bacterial DNA was extracted with QIAamp minikit (Qiagen) and digested with MmeI enzyme for 4 hours at 37°C. Samples were then purified with QIAquick PCR Purification kit (Qiagen) and run on 2% agarose gel to only retrieve fragments between 1500 and 2000 bp, which were then purified with QIAquick Gel Extraction kit (Qiagen). DNA was next ligated to Illumina adapters and amplified by PCR to obtain libraries, which were then sequenced using a NextSeq500.

## Analysis pre-processing steps

Reads were collected in a single file per sequencing run. The first step was to filter out PhiX reads, PhiX being an Illumina technology control. Reads were mapped against the PhiX sequence using bowtie2 [22] and the resulting unmapped reads were retained. PhiX reads accounted for 24% and 23% of the total of reads in each run, respectively. The reads contained the P2 adapter, followed by the sample-specific barcode (6 bp), 16–18 bp of the PA genome, and then the reverse complement of the 30-nucleotide P1M6GAMmeI sequence (ACAGGTTGGATGATAAGTCCCCGGTCTATC). To identify the reads from each sample and extract the 16–18 bp of PA for downstream analysis, cutadapt [23] was used with this command line: cutadapt -g "sample_BARCODE(6nt);min_overlap=6;e=0...ACAGGTTGGATGATAAGTCCCCGGTCTATC;min_overlap=30;e=0" --discard-untrimmed -m 16 -M 18 -j 0 -o output.fastq noPhiX.fastq.

This command allows for the recognition of the 6-nucleotide sample-specific barcode followed by the transposon tag sequence (ACAGGTTGGATGATAAGTCCCCGGTCTATC), requiring a perfect match (e=0) with a minimum overlap of 30 bp. Only reads containing both the barcode and the transposon tag were retained (--discard-untrimmed), and sequences between 16 and 18 bp were extracted for further analysis. At the end of this step, a Fastq file was obtained for each sample containing only the 16–18 bp sequences from PA genome. These Fastq files have been deposited in the Sequence Read Archive (SRA) (www.ncbi.nlm.nih.gov/sra) under accession number PRJNA1240204.

## Read mapping and reads-counts files

Two mappers were used in this study using the trimmed fastq files: BWA [19] and bowtie [18]. BWA version 0.7.12 was directly used with the TRANSIT2 v1.1.7 tpp command on the PA UCBPP-PA14 genome sequence (NC_008463.1), producing a.wig file. Bowtie 1.1.2 was used with the option -v 0 (no mismatches) -m 1 (report only unique mapping) producing a sam file. Scripts from https://github.com/SuzanneWalkerLab/5SATnSeq (sam_to_tabular.py and make_wig.py) were used to obtain a.wig file from the alignment file. FiTnEss took a sam file as input and returned a tally file with the script "tally.py".

## Methods to detect essential genes

TRANSIT2 (v1.1.7) was used. To evaluate the impact of the mapper, the Gumbel, HMM and TTNfitness functions were used with default parameters, except for HMM, where the "-conf-on" parameter was enabled to obtain additional columns with confidence information.

In the section evaluating the impact of the statistical methods, The Gumbel function was used with default parameters. The HMM function was used with the "--n quantile -l -conf-on" options. FitNess function was run in R for each sample, with default parameters using the tally file in input. We manually updated the reference genome to NC_008463.1. For each sample, genes with an adjusted p-value <0.05 were considered as essential. The lists of EGs from each replicate of a given condition were intersected to obtain the EGs of this condition. Upset plots were generated using the UpSetR package in R [24].

## Methods to determine conditional gene essentiality

TRANSIT2 (v1.1.7) was used with default parameters for statistical methods: resampling, utest, ANOVA and ZINB. For intersecting methods, the lists of EGs were taken from the part "Impact of statical method on providing sets of essential genes", for HMM, the list corresponded to HMM_GD, which considers both essential and GD genes, and FiTnEss_FDR for FDR. Then, the lists of EGs corresponding to the PA14 WT and PA14 Δ*oprD* conditions were intersected, and only the genes specific to one condition or the other were kept.

## EGs annotation

All the genes were annotated and clustered in function class using available resources from different databases like BioCyc Genome Database Collection (https://biocyc.org/PAER208963/organism-summary), The *Pseudomonas* Genome Database (https://Pseudomonas.com/) [25] or GenomeNet (https://www.genome.jp/dbget-bin/www_bget?gn:pau).

## Deletion mutant construction

Deletion of *shaF* gene in PA14 WT and Δ*oprD* strains were performed using the replacement vector pEXG2. Upstream and downstream fragments of ~ 500pb flanking the *shaF* gene were amplified by PCR from PA14 genomic DNA using overlapping primers. Linearized fragment of pEXG2 was also obtained by PCR. The 3 fragments were ligated using the Gibson Assembly Cloning Kit (New England Biolabs). The created pEXG2::*shaF* was transformed into *E. coli* DH5α and positive clones were selected on LB agar plates containing 15 µg/mL of gentamicin. To transfer the plasmid in PA14 strains, triparental conjugations were performed between the donor strain *E. coli*/pEXG2::*shaF*, the recipient strain PA14 WT or Δ*oprD* and the helper strain *E. coli* HB101/pRK2013. The PA14 transconjugants were selected using LB agar plates supplemented with irgasan 25 µg/mL and gentamicin 75 µg/mL. The merodiploid gentamicin-resistant PA14 strain was then cultured in LB broth containing 75 µg/mL of gentamicin to exponential growth and streaked onto LB agar plates containing 18% sucrose for the allelic exchange. Sucrose-resistant colonies were tested to confirm gentamicin susceptibility, indicating excision from the genome of the pEXG2 backbone by double-cross-over event and thus gene replacement. The *shaF* gene deletion was then confirmed by PCR and sequencing. The plasmids and primers used were listed in S3 Table.

## Supporting information

**S1 Fig. Impact of normalization in determining essential genes. (A)** Number of EGs identified with the HMM method and seven types of normalization, considering either only essential category or essential plus Growth Defect categories [Es + GD]. The proportions of low-confidence and ambiguous categories are shown. **(B)** Fold-enrichment of gold-standard genes recovered by each method (bars), with statistical significance indicated by p-value codes

(*** $p < 1 \times 10^{-10}$, **$p < 1 \times 10^{-5}$, *$p < 1 \times 10^{-3}$, ns = not significant). Recall values are shown as diamond markers on the secondary axis. Colors correspond to the gold-standard datasets and the strain (WT or delta); in red the set containing the 84 core EGs for PA14 WT, in blue the set containing the 115 gold-standard genes for PA14 WT and in green the set containing the 84 core EGs for PA14 ΔoprD.
(DOCX)

**S1 Table. Impact of parameters on providing sets of essential genes.**
(DOCX)

**S1 Text. Impact of parameters on providing sets of essential genes.**
(DOCX)

**S2 Fig. Upset plot displaying the intersections of the essential genes sets obtained from the seven normalizations with the HMM method of TRANSIT2 for the WT condition.** Each vertical bar represents a distinct set of genes, and the plot highlights the overlapping gene sets.
(DOCX)

**S3 Fig. Upset plot displaying the intersections of the essential and growth defect genes sets obtained from the seven normalizations with the HMM method of TRANSIT2 for the WT condition.** Each vertical bar represents a distinct set of genes, and the plot highlights the overlapping gene sets.
(DOCX)

**S4 Fig. Upset plot displaying the intersections of the essential genes sets obtained from the seven normalizations with the HMM method of TRANSIT2 for the PA14ΔoprD condition.** Each vertical bar represents a distinct set of genes, and the plot highlights the overlapping gene sets.
(DOCX)

**S5 Fig. Upset plot displaying the intersections of the essential and growth defect genes sets obtained from the seven normalizations with the HMM method of TRANSIT2 for the PA14ΔoprD condition.** Each vertical bar represents a distinct set of genes, and the plot highlights the overlapping gene sets.
(DOCX)

**S2 Text. Impact of normalization on identifying essential genes.**
(DOCX)

**S6 Fig. Upset plot displaying the intersections of the essential genes sets obtained from different statistical methods (Gumbel, HMM_GD and FiTnEss_FDR) for the PA14WT condition.** Each vertical bar represents a distinct set of genes, and the plot highlights the overlapping gene sets.
(DOCX)

**S7 Fig. Upset plot displaying the intersections of the essential genes sets obtained from different statistical methods (Gumbel, HMM_GD and FiTnEss_FDR) for the PA14ΔoprD condition.** Each vertical bar represents a distinct set of genes, and the plot highlights the overlapping gene sets.
(DOCX)

**S2 Table. Gold Standard List_GO.**
(XLSX)

**S3 Table. Plasmids and primers used.**
(DOCX)

**S4 Table. List of abbreviation.**
(DOCX)

## Acknowledgments

We would like to deeply thank Delphine Beury, Dr David Hot and Dr Christophe Audebert, for their advice and technical assistance.

## Author contributions

**Conceptualization:** Anaëlle Muggeo, Thomas Guillard.

**Data curation:** Cléophée Van Maele, Ségolène Caboche.

**Formal analysis:** Cléophée Van Maele, Ségolène Caboche, Nathan Nicolau-Guillaumet.

**Validation:** Anaëlle Muggeo, Thomas Guillard.

**Writing – original draft:** Cléophée Van Maele, Ségolène Caboche, Thomas Guillard.

**Writing – review & editing:** Cléophée Van Maele, Ségolène Caboche, Nathan Nicolau-Guillaumet, Anaëlle Muggeo, Thomas Guillard.

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
