## [Decision Letter · Decision Letter 0]

28 Aug 2025

Introducing gold-standard essential gene datasets for Pseudomonas aeruginosa to enhance Tn-Seq analyses

PLOS Computational Biology

Dear Dr. Guillard,

Thank you for submitting your manuscript to PLOS Computational Biology. After careful consideration, we feel that it has merit but does not fully meet PLOS Computational Biology's publication criteria as it currently stands. Therefore, we invite you to submit a revised version of the manuscript that addresses the points raised during the review process.

Please submit your revised manuscript within 60 days Oct 28 2025 11:59PM. If you will need more time than this to complete your revisions, please reply to this message or contact the journal office at ploscompbiol@plos.org. Please include the following items when submitting your revised manuscript:

We look forward to receiving your revised manuscript.

Kind regards,

Eric C. Dykeman, Ph.D.

Academic Editor

PLOS Computational Biology

Dominik Wodarz

Section Editor

PLOS Computational Biology

**Additional Editor Comments (if provided):**

Dear Authors,

Please pay close attention to the comments of reviewers 2 and 3 when drafting your revision and response.

Kind regards.

**Journal Requirements:**

**Reviewers' comments:**

Reviewer's Responses to Questions

**Comments to the Authors:**

Reviewer #1: Defining the essential genes during TnSeq experiments is a mandatory step. Once these genes defined, the selective pressure chosen by the researchers to be applied on the saturated bank of mutants will define the genes important or essential for survival in the different conditions chosen by the scientists.

As of today, no state of the art methodology has been published for finding and defining these essential genes.

Major Comments:

Can groups mapping and analyzing their TnSeq data with CLC genomics use the authors approach and if yes, how?

A final schematic to summarize their approach

In the discussion, an additional emphasize on how this extensive work will be used by other researchers could be useful

Minor comments:

1) More explanation for the use of the Delta-oprD strain could be helpful for the reader

2) Line 169: remove the (…), be more specific or add a reference

Reviewer #2: The authors performed a systematic study of commonly used analysis tools for Tn-Seq, TRANSIT2 and FiTnEss, from which they offer suggestions for parameter settings. They also provide two lists of essential genes in P. aeruginosa that they termed to be “gold-standard” sets – one specific for the strain PA14 for grown in LB and the other applicable to any Tn-seq experiment in P. aeruginosa. The authors further generated their own Tn-Seq datasets from wild-type PA14 and an OprD-deficient strain, which they used to test the parameter settings of the various tools. While the suggestions for parameters to use for analysis would be helpful to other researchers, there are crucial issues that need to be addressed. First, the gold-standard set that they delineate is not much different than what has been previously published in the literature (Poulsen et al.) and the choices made in creating those gene sets were not well-defined. It is not clear that they are appropriate gene sets for comparing analytical tools. Moreover, the authors do not provide a comprehensive metric with which to evaluate essential gene lists, as they only consider how many of the gold-standard genes are represented, but do not have any assessment of potential false positives or false negatives. Last, there are other potential biases that need to be addressed more thoroughly and are detailed in the comments below.

Major comments

1.The “gold-standard” set of essential genes is very similar to what was already published in Poulsen et al. (Dataset S6), in intersecting gene lists from the same previous studies as utilized in this manuscript (Table 2). There is not enough technical justification as to why the gold standard sets defined in the manuscript should be termed “gold standard.” The GOLD_84 set particularly seems arbitrary, in that many sets are already intersections of one another, not all possible intersections are considered (e.g. Poulsen CORE could be defined using FDR and/or FWER thresholds), and some sets have more confidence than others (e.g. Poulsen FWER and FDR). Thus, the term “gold standard” is not well-defined and is overselling the novelty of the gene sets. The manuscript needs justification for the suitability of the gene sets for comparing analysis methods by providing explanations for why the authors chose to intersect the gene sets that they used and why they chose to take an intersection rather than a union of gene sets.

2.The only metric the authors provide to evaluate the Tn-Seq tools is the percentage of gold-standard genes that are in the essential gene list. However, that metric does not consider how many genes are present in the essential gene list, which may be false positives. For example, if a tool outputs all the genes in the genome as its list of essential genes, it would capture 100% of the gold-standard genes, and would be “better” than the tools listed in this study. There is no penalty for the number of false positives given by the tool. The authors need to provide a more robust metric (or metrics) to quantify the performance of the tools with the given parameters, accounting for potential false positives, and possibly even false negatives.

3.The authors mention multiple other tools that are used in Tn-Seq analysis like Magenta, TSAS, etc but restrict themselves to only testing TRANSIT2 and FiTnEss. For a more comprehensive comparison of different methods, it would be ideal to test all available methods with different parameter settings. This could also give more insight into the reliability of the ‘gold-standard’ sets as a metric for determining the performance of a tool.

4.The section “Impact of the mapper on identifying essential genes” has potential biases that need to be addressed. Specifically, the mapper used in the previous studies were not stated, and it is possible that bowtie yielded more of the gold standard genes than BWA simply because all the studies from which the gold standard genes were derived also used bowtie rather than BWA. The result may not be due to bias in the data rather than to actual differences in the mappers. Could the authors add which aligners were used in each of the studies in Table 2, and adjust their interpretation accordingly?

5.Also in the section “Impact of the mapper on identifying essential genes,” the authors should more comprehensively assess whether normalization techniques could adjust for differences in performance between BWA and bowtie. In Table 3, BWA seems superior to bowtie in most metrics except for skewness, and yet the authors suggest to use bowtie. Would it be possible that using alternate normalization techniques such as the “betageom” option in TRANSIT2 could account for the skewness and improve the performance using BWA? It seems that the aligner was chosen prior to adjusting any of the other parameters in the TRANSIT2 tool, and that the testing was not comprehensive in this regard.

6.While the authors provide empirical evidence for some parameters yielding better results than others, they do not provide explanations to explain why those results occur. Could authors provide some technical conjectures as to why the parameters they chose yielded the best results?

Minor comments

1.In Figure 3b, is the delta-oprD for the 115 gene set missing?

2.In the section comparing bwa and bowtie, the lower bound for the density metric to assess performance of the aligner is given as 35%, while the citation used for the same gives it as 30%.

3.The language in the “Methods to determine conditional gene essentiality” section needs to be clarified. Particularly when intersections are mentioned, it would be helpful to specifically state what the intersection is between. Also, it would help to state explicitly that the genes that were not in the intersection between the essential genes in wild-type versus delta-oprD were considered to be the conditionally essential genes. Currently it seems to read that the genes in the intersection were the conditionally essential genes.

4.At the end of the section “Methods to determine conditional gene essentiality,” it states “we propose FiTnEss_FDR complemented with information retrieved from ZINB, would be an accurate and reliable method…” Could the authors explain how they came to this conclusion and practically how a researcher would “complement the information” with the other method?

5.It would be helpful to have a table describing all the method abbreviations used in the manuscript. It would also be helpful to be consistent in referring to these abbreviations in both the figures and the text. For example, sometimes it seems that HMM is used interchangeably with HMM_GD, which can be confusing to the reader.

Reviewer #3: This study by Maele et al. addresses a critical need in the Pseudomonas aeruginosa research community by establishing gold-standard essential gene (EG) datasets to benchmark bioinformatics pipelines for transposon sequencing (Tn-Seq) analyses. Drawing on both literature and newly generated Tn-seq data from PA14 wild-type and ΔoprD strains grown in LB, the authors compared EG lists produced by several statistical methods implemented in TRANSIT2 and FiTnEss. They constructed a reference set of 84 EGs for P. aeruginosa, as well as a PA14-LB-specific list of 115 EGs, and assessed how effectively these genes could be identified using their datasets. Retrieval rates varied across methods, with the Hidden-Markov Model in TRANSIT2 recovering approximately 90% of gold-standard EGs and FiTnEss achieving up to 100%. These curated datasets will prove valuable for the Tn-seq community, providing objective benchmarks for evaluating analysis pipelines, thereby improving reproducibility, standardization, and essential gene identification. The study is well organized, with clear writing and effective figures. I have several suggestions to enhance clarity and expand the scope of the work.

Major comments:

1. The motivation for utilizing the ΔoprD mutant in Tn-seq experiments is not clear. Further, the subsequent comparison of statistical tests for identifying conditional essential genes between the wild type and ΔoprD backgrounds is somewhat confusing. Could the authors include a known essential gene that displays conditional essentiality in the ΔoprD background (but not in wild type) as a positive control to validate the statistical tests? Alternatively, have the authors performed experimental validation—for instance, by comparing gene deletions in both backgrounds, such as with shaF? Additionally, what would the authors predict if a similar comparison was conducted in another gene deletion background?

2. The authors refer to two essential gene lists from prior studies: GOLD 84 (general P. aeruginosa essential genes) and GOLD 115 (PA14-specific essential genes). While it is noted that the gene ontology profiles of these lists are similar, it would be helpful if the authors could highlight and discuss the key differences between these two sets to provide readers with more context.

3. For these two essential gene lists, it would be useful to include a summary of the mappers and statistical approaches employed in prior studies in a table. It would be important for the authors to clarify how robust the ‘gold standard’ lists are to different analysis methods.

4. The focus on PA14 is understandable given the availability of high-quality Tn-seq datasets, but this may limit the generalizability of the findings. Could the authors consider extending their analysis to PAO1 using publicly available Tn-seq datasets or discuss the potential for broader application of their evaluation framework to non-Pseudomonas Tn-seq studies?

Minor comments:

1. The resolution of Figure 1A and 1B is low and should be improved for clarity.

**Have the authors made all data and (if applicable) computational code underlying the findings in their manuscript fully available?**

Reviewer #1: Yes

Reviewer #2: Yes

Reviewer #3: Yes

PLOS authors have the option to publish the peer review history of their article (what does this mean? ). If published, this will include your full peer review and any attached files.

**Do you want your identity to be public for this peer review?** For information about this choice, including consent withdrawal, please see our Privacy Policy .

Reviewer #1: No

Reviewer #2: No

Reviewer #3: No

**Figure resubmission:**

**Reproducibility:**



---

## [Decision Letter · Decision Letter 1]

25 Jan 2026

Dear Prof Guillard,

We are pleased to inform you that your manuscript 'Introducing gold-standard essential gene datasets for Pseudomonas aeruginosa to enhance Tn-Seq analyses' has been provisionally accepted for publication in PLOS Computational Biology.

Best regards,

Eric C. Dykeman, Ph.D.

Academic Editor

PLOS Computational Biology

Dominik Wodarz

Section Editor

PLOS Computational Biology

Reviewer's Responses to Questions

**Comments to the Authors:**

Reviewer #1: No more comments

Reviewer #3: The authors have answered most of my comments. This study will serve as a useful benchmark for the P. aeruginosa community.

**Have the authors made all data and (if applicable) computational code underlying the findings in their manuscript fully available?**

Reviewer #1: Yes

Reviewer #3: Yes

PLOS authors have the option to publish the peer review history of their article (what does this mean? ). If published, this will include your full peer review and any attached files.

**Do you want your identity to be public for this peer review?** For information about this choice, including consent withdrawal, please see our Privacy Policy .

Reviewer #1: No

Reviewer #3: No

---

## [Editor Report · Acceptance letter]

PCOMPBIOL-D-25-01258R1

Introducing gold-standard essential gene datasets for Pseudomonas aeruginosa to enhance Tn-Seq analyses

Dear Dr Guillard,

I am pleased to inform you that your manuscript has been formally accepted for publication in PLOS Computational Biology. Your manuscript is now with our production department and you will be notified of the publication date in due course.

With kind regards,

Aiswarya Satheesan
